# Effect of the Ketone Body, D-β-Hydroxybutyrate, on Sirtuin2-Mediated Regulation of Mitochondrial Quality Control and the Autophagy–Lysosomal Pathway

**DOI:** 10.3390/cells12030486

**Published:** 2023-02-02

**Authors:** Juan Carlos Gómora-García, Teresa Montiel, Melanie Hüttenrauch, Ashley Salcido-Gómez, Lizbeth García-Velázquez, Yazmin Ramiro-Cortés, Juan Carlos Gomora, Susana Castro-Obregón, Lourdes Massieu

**Affiliations:** 1Department of Molecular Neuropathology, Instituto de Fisiología Celular, Universidad Nacional Autónoma de México (UNAM), México CP04510, Mexico; 2Deparment of Neurodevelopment and Physiology, Instituto de Fisiología Celular, Universidad Nacional Autónoma de México (UNAM), México CP04510, Mexico

**Keywords:** autophagy, ketone bodies, lysosomal biogenesis, mitophagy, sirtuin2

## Abstract

Mitochondrial activity and quality control are essential for neuronal homeostasis as neurons rely on glucose oxidative metabolism. The ketone body, D-β-hydroxybutyrate (D-BHB), is metabolized to acetyl-CoA in brain mitochondria and used as an energy fuel alternative to glucose. We have previously reported that D-BHB sustains ATP production and stimulates the autophagic flux under glucose deprivation in neurons; however, the effects of D-BHB on mitochondrial turnover under physiological conditions are still unknown. Sirtuins (SIRTs) are NAD^+^-activated protein deacetylases involved in the regulation of mitochondrial biogenesis and mitophagy through the activation of transcription factors FOXO1, FOXO3a, TFEB and PGC1α coactivator. Here, we aimed to investigate the effect of D-BHB on mitochondrial turnover in cultured neurons and the mechanisms involved. Results show that D-BHB increased mitochondrial membrane potential and regulated the NAD^+^/NADH ratio. D-BHB enhanced FOXO1, FOXO3a and PGC1α nuclear levels in an SIRT2-dependent manner and stimulated autophagy, mitophagy and mitochondrial biogenesis. These effects increased neuronal resistance to energy stress. D-BHB also stimulated the autophagic–lysosomal pathway through AMPK activation and TFEB-mediated lysosomal biogenesis. Upregulation of SIRT2, FOXOs, PGC1α and TFEB was confirmed in the brain of ketogenic diet (KD)-treated mice. Altogether, the results identify SIRT2, for the first time, as a target of D-BHB in neurons, which is involved in the regulation of autophagy/mitophagy and mitochondrial quality control.

## 1. Introduction

Neurons rely on glucose oxidative metabolism through the tricarboxylic acid cycle (TCA) and oxidative phosphorylation to produce ATP and support their high energy requirements [1,2]. Thus, mitochondria are essential organelles for neuronal homeostasis. As post-mitotic cells, neurons cannot renew their mitochondria through cell division; therefore, mitochondrial quality control relies on mitochondrial fusion and fission, biogenesis and mitophagy, a specific autophagic pathway underlying the removal of dysfunctional or damaged mitochondria [3]. Accumulating evidence supports that mitophagy declines during aging and neurodegeneration [3,4], but the role of mitophagy on mitochondrial turnover under physiological conditions is still not well understood. The mammalian target of Rapamycin (mTOR) and AMP-activated protein kinase (AMPK) are key regulators of autophagy initiation and autophagosome formation. Activation of mTOR by the PI3K/AKT pathway leads to the phosphorylation of ULK1 S757 and hence to the inhibition of the autophagy initiation complex. Meanwhile, AMPK inactivates mTOR through the phosphorylation of RAPTOR, which reduces the phosphorylation of ULK1 S757 by mTOR. Conversely, AMPK can initiate autophagy through ULK1 S317 phosphorylation [5,6].

Alternative substrates to glucose, such as the ketone bodies (KB), acetoacetate and β-hydroxybutyrate (BHB), can sustain ATP production and neuronal function during glucose shortage conditions. In previous studies, we observed that D-BHB prevents ATP decline and reduces reactive oxygen species production and neuronal death in cultured neurons exposed to glucose deprivation (GD) and the brain of hypoglycemic rats [7,8]. More recently, we reported that BHB exposure restores the impaired autophagic flux in vivo and in vitro under glucose-limiting conditions, although the mechanisms involved have not been elucidated [9,10].

Due to its multiple beneficial effects, the use of the ketogenic diet (KD) or the exogenous administration of BHB has been suggested for the treatment of acute brain injury induced by brain trauma and stroke and chronic neurodegenerative diseases such as Alzheimer’s and Parkinson’s [11]. However, little is known about the effect of these treatments on the regulation of autophagy and mitophagy in the brain of healthy adult animals or neurons under normal culturing conditions.

The protective effects of BHB have been attributed to its metabolic action after its conversion to acetyl-CoA and ATP production through the TCA and oxidative phosphorylation [12]. However, in the last decade, diverse actions of BHB have been discovered, which can also account for its protective effects. For example, through inhibition of histone-deacetylases (HDACS) and β-hydroxybutyrylation of histones and other proteins, BHB can regulate gene expression [13,14], it act as an anti-inflammatory [15] and as an antioxidant molecule [7] and signal through G-protein coupled receptors (GPCRs), including GRP41 and the hydroxy-carboxylic acid receptor 2, HCAR_2_/GPR109A [16,17]. In addition, KB metabolism increases the NAD^+^/NADH ratio [18,19], favoring the activation of NAD^+^-dependent enzymes such as sirtuins (SIRTs), a class III histone deacetylases (Class III HDACS). NAD^+^ is a co-substrate of SIRTs, which have been involved in metabolic diseases and aging [20]. SIRTs are important regulators of autophagy and mitophagy through the deacetylation of FOXO1 and FOXO3a transcription factors, and they regulate mitochondrial and lysosomal biogenesis by deacetylation of PGC1α and TFEB, respectively [21,22,23,24,25].

In the present study, we investigated the effects of D-BHB on basal autophagy, mitophagy and mitochondrial and lysosomal biogenesis in healthy cortical cultured neurons. Results show that D-BHB exposure improves mitochondrial function and stimulates autophagy, mitophagy and mitochondrial biogenesis through the upregulation of the transcription factors FOXO1 and FOXO3a and the transcription coactivator PGC1α in a SIRT2-dependent manner. D-BHB also stimulates AMPK activity contributing to autophagy initiation and enhanced lysosomal biogenesis by TFEB upregulation. A two-week ketogenic diet in mice augmented D-BHB blood levels and increased PGC1α, FOXO1, FOXO3a and TFEB abundance in the brain, supporting that D-BHB stimulates autophagy and lysosomal/mitochondrial biogenesis in neurons, which can improve mitochondrial quality control.

## 2. Materials and Methods

### 2.1. Cell Culturing

Neuronal cultures from rat cerebral cortex were obtained from E17 embryos. Animals were treated following the *National Institute of Health Guide for the Care and Use of Laboratory Animals* (NIH publications N0. 80–23, revised 1996). The protocol for euthanasia was approved by the Animal Care Committee (CICUAL, LMT160-20) of the Instituto de Fisiología Celular, UNAM, and only the necessary number of animals was used. Briefly, cerebral cortex was dissected, chopped and incubated for 3 min with 0.25% trypsin/10% EDTA solution at 37 °C; Soybean trypsin inhibitor (0.52%) and DNase (0.08%) were used to stop digestion. Cells were suspended in Neurobasal medium (Gibco, 21103-049, Grand Island, NY, USA) supplemented with 1% B27 containing antioxidants (Gibco, 17504-044), 1% B27 with no antioxidants (Gibco, 10889-038), 0.5 mM L-Glutamine and 20 mg/mL gentamycin (Gibco, 15710-064). Cells were seeded in poly-L-Lysine (Sigma-Aldrich, P-1524, St. Louis, MO, USA) pre-coated wells at a density of 2.2 × 10^5^ cells/cm^2^. Cells were maintained at 37 °C a humidified 5% CO_2_/95% air atmosphere for 8 days in vitro (DIV), and cytosine-D-Arabinoside (0.54  μM, Sigma-Aldrich, C-1768) was added after 4 DIV.

### 2.2. Cell Treatments

After 8 DIV neurons were incubated with the following compounds: 6 to 48 h with D-β-hydroxybutyrate (5 mM; Sigma-Aldrich, 298360-1G), 24 h with L-β-hydroxybutyrate (5 mM; Sigma-Aldrich, H3145-1G), 24 h with β-NAD (5 mM; Sigma-Aldrich, N-7004), 4 h with chloroquine (20 µM; Sigma-Aldrich, C6628-25G) and 24 h with AGK2 (SIRT2 inhibitor, 5 µM; Merk Millipore, 566324-5MG, Burlington, MA, USA).

### 2.3. qRT-PCR

Trizol (Invitrogen, 15596018, Waltham, MA, USA) was used to extract total RNA. The High Capacity cDNA Reverse Transcription Kit (Applied Biosystems, 4368814, Waltham, MA, USA) was used to synthesize cDNA from 2 µg RNA. Reverse transcription reaction was carried out in a thermal cycler as follows: 25 °C for 10 min, 37 °C for 120 min and 85 °C for 5 min. We used Maxima SYBR green qPCR Master Mix (Thermo Fisher Scientific, K0252, Waltham, MA, USA) for real-time PCR using 50 ng of cDNA for each reaction. The primers used are shown in Appendix A.

### 2.4. MTT Cell Viability Assay

Cortical neurons viability was measured by the 3-(4, 5-dimethylthiazol-2-yl)-2, 5-diphenyltetrazolium bromide (MTT, Sigma-Aldrich, M2128) reduction assay, which is an index of mitochondrial dehydrogenase activity. MTT (60 μg/mL) was added at the end of treatments and incubated for 1 h at 37 °C. The medium was removed, and 1 mL isopropanol-HCl was added to solubilize the resulting formazan salt. Formazan absorbance was determined at 570 nm in a Beckman Coulter life science UV/Vis spectrophotometer. Data are expressed as percent of MTT reduction relative to control.

### 2.5. Lactate Dehydrogenase (LDH) Activity

Culture medium was withdrawn, and lactate dehydrogenase (LDH) activity was determined following the decrease in the optical density resulting from the oxidation of NADH using pyruvate as a substrate. At the end of treatments, 200 μL of culture medium was collected and incubated with 9.4 mM NADH in 1 mM K_2_HPO_4_/KH_2_PO_4_ buffer for 5 min at room temperature. Pyruvate (20 mM) (Sigma-Aldrich, P-2256) was used to start the reaction, and the change in optical density at 340 nm was followed after 5 min in a Beckman Coulter life science UV/Vis spectrophotometer. Data are expressed as percent LDH release relative to control.

### 2.6. Subcellular Fractionation

Cells were cultured in 60 mm dishes, and at the end of treatments, they were washed with ice-cold 0.1 M PBS and scraped in buffer A containing: 0.25 M Sucrose, 1 mM EDTA, 20 mM HEPES, 10 mM KCl, 1.5 mM MgCl_2_, 0.1% Triton X-100, 1 mM DTT and 2 mg/mL of protease inhibitor cocktail (cOmplete, Roche, 11836145001, Mannheim, Germany). The mixture was gently shaken for 10 min at 4 °C and centrifuged at 1000 g for 10 min. The supernatant was collected as the cytoplasmic extract. The pellet was washed with buffer A and re-suspended in lysis buffer (50 mM Tris-HCl pH 8.0, 150 mM NaCl, 1% Triton X-100, 0.5% sodium deoxycholate, 1% SDS) containing 2 mg/mL of protease inhibitor cocktail (cOmplete, Roche, 11836145001). This was used as the nuclear fraction.

### 2.7. Immunoblotting

Cells were cultured on 35 mm dishes. At the end of treatments, they were washed with ice-cold 0.1 M PBS and lysed with lysis buffer (50 mM Tris-HCl pH 8.0, 150 mM NaCl, 1% Triton X-100, 0.5% sodium deoxycholate, 1% SDS) containing 2 mg/mL of protease inhibitor cocktail (cOmplete, Roche, 11836145001). After, cell lysates were centrifuged at 1500 g at 4 °C for 5 min. The Lowry method was used to determine protein concentration, and 30 μg of protein from each sample was separated in SDS-PAGE and then transferred to a 0.45 μm PVDF membrane (Immobilon-P Membrane, Merck Millipore, IPVH00010). One hour incubation at room temperature with dry milk (5%) in TBS buffer (50 mM Tris-HCl pH 7.5, 150 mM NaCl and 0.1% Tween 20) was used to block the membranes, and primary antibodies were incubated overnight at 4 °C and for 1 h at room temperature. The following primary antibodies and dilutions were used: FOXO1, 1:3000 (Cell Signaling, 2880S, Danvers, MA, USA); FOXO3a, 1:1000 (Cell Signaling, 2497S); TFEB, 1:1000 (Abcam, ab2636, Cambridge, UK); PGC-1α, 1:500 (Santa Cruz Biotechnology, sc-5816, Dallas, TX, USA); LC3, 1:6000 (MBL International, PD014, Woburn, MA, USA); SQSTM1/p62, 1: 2000 (Cell Signaling, 5114S); TOM20, 1:8000 (Cell Signaling, 42406S); COX IV, 1:3000 (Cell Signaling, 11967S); CTSB, 1:3000 (Sigma-Aldrich, C6243); LAMP2, 1:7000 (Sigma-Aldrich, L0668), SIRT2, 1:1000 (Sigma-Aldrich, S5313); Parkin, 1:3000 (Cell Signaling, 4211S); Bnip3, 1:500 (Santa Cruz Biotechnology, sc-56167); NIX/Bnip3L, 1:3000 (Cell Signaling, 12396S); H1.0, 1:10000 (Abcam, ab11079); GAPDH, 1:16000 (Cell Signaling, 2118); Actin, 1:8000 (Merck Millipore, MAB1501). For the detection of primary antibodies, a secondary anti-rabbit or anti-mouse goat antibody coupled to peroxidase was used, and immunoreactivity was detected using the chemiluminescent HRP substrate (Merck Millipore, WBLUF0500). C-DiGit Scanner (LI-COR) was used to develop the membranes and the ImageJ software version 2.1.0 (Wayne Rasband (NIH), USA) to analyze images.

### 2.8. Transfection of Neurons

Neurons were cultured on 18 mm diameter coverslips pre-coated with poly-L-Lysine (Sigma-Aldrich, P-1524), and at 4 DIV, they were transfected with 2 µg of plasmid DNA per coverslip using Lipofectamine 2000 (Thermo Fisher Scientific, 11668-019) or Lipofectamine LTX (Thermo Fisher Scientific, 11668-019). A 1:3 ratio of µg plasmid DNA and µL lipofectamine was used. Neurons were incubated with the transfection mixture for 24 h. Plasmid pmRFP-LC3 (Addgene, #21075, Watertown, MA, USA) kindly provided by Dr. Susana Castro Obregón and mKeima-red-mito-7 (Addgene, #56018) were transfected.

### 2.9. Immunofluorescence

Neurons were cultured on 18 mm diameter coverslips pre-coated with poly-L-Lysine (Sigma-Aldrich, P-1524). At the end of treatments, cells were washed with 0.1 M PBS and incubated for 20 min with 4% paraformaldehyde at room temperature. Coverslips were incubated with 5% Albumin-PBS and 0.1% Triton X-100 for 1 h at room temperature. Anti-TOM20 (1:250, Cell Signaling, D8T4N) or anti-SIRT2 (1:200, Sigma-Aldrich, S5313) antibody was added overnight at 4 °C, and then Alexa 488 anti-rabbit antibody (Jackson Immunoresearch Laboratories, West Grove, PA, USA) was used. Finally, cell nuclei were stained with 0.001% Hoechst (Sigma-Aldrich, 33258) in PBS, and coverslips were mounted with Fluoromount-G^TM^ (Electron Microscopy Sciences, 17984-25, Hatfield, PA, USA).

### 2.10. Mitophagy Analysis

Images of seven different transfected neurons with RFP-LC3 per coverslip and immunofluorescence against TOM20 were acquired using an inverted Zeiss LSM 800 confocal microscope and a Plan-Apochromat 63x/1.4 NA objective. Transfected neurons were identified by the presence of red fluorescent dots. RFP-LC3 was excited with a 561 nm laser, and TOM20 was observed with a 488 nm laser. The maximum projection of each image was obtained from a z-stack and analyzed with the CellProfiler program. Cells were delineated, and the number of RFP-LC3-positive particles (range size 0.3–1.5 µM) per cell was quantified. After detection of the RFP-LC3 particles, the colocalization of each particle with TOM20 immunoreactivity (green fluorescence) was determined using Pearson’s coefficient. Those particles with a coefficient greater than 0.5 were taken as positive colocalizing particles. Additionally, the area of these particles was measured, and the total area of colocalizing particles per cell was measured.

### 2.11. Mitochondrial Membrane Potential Analysis

Neurons were cultured on 35 mm glass bottom dishes pre-coated with poly-L-Lysine (Sigma-Aldrich, P-1524) for 8 DIV. After treatments, cells were incubated with 1 μM JC-1 dye for 20 min. Then, living cells were maintained at 37 °C in a humidified 5% CO_2_/95% air atmosphere and observed by an inverted Zeiss LSM 800 confocal microscope using a Plan-Apochromat 63x/1.4 NA objective. JC-1 monomer was excited at 488 nm, and JC-1 aggregates were excited at 561 nm. Five images were taken and analyzed per condition. The maximum projection of each image was obtained from a 7 μm-thick z-stack and analyzed with CellProfiler program. Green and red fluorescence were measured, and results are expressed as the red/green fluorescence ratio.

### 2.12. ATP Determination

Cells were plated on 12-well plates for 8 DIV. After treatments, ATP levels were determined with the luciferin–luciferase quimioluminescent kit (Molecular Probes, Eugene, OR, USA), as previously described [8]. A standard curve (from 6.5 to 250 pmol) was used to calculate ATP concentrations. Protein was determined by Lowry’s method from aliquots of cell homogenates, and data are expressed as pmol of ATP/μg of protein.

### 2.13. Mitochondrial Morphology Analysis

Seven transfected neurons with mKeima-red-mito-7 (mt-Keima), identified by the presence of red fluorescent mitochondria, were analyzed per coverslip. Images were taken using an inverted Zeiss LSM 800 confocal microscope and a Plan-Apochromat 63x/1.4 NA objective. The maximum projection of each image was obtained from a z-stack and analyzed with the CellProfiler program. Particles were segmented based on their diameter (range size 0.4–15 µm), and particles above 6 µm^2^ were excluded. Then, mitochondrial area and maximal Feret´s diameter were measured.

### 2.14. Autofluorescence of NADH

After treatment, cells cultured in 35 mm dishes were observed using a two-photon microscope (LSM 710, Carl Zeiss, Jena, Germany) equipped with a Chameleon Ultra II pulsed infrared laser (Coherent Inc., Santa Clara CA, USA), and a W-Apochromat 63x/1.0 NA objective was used. NADH was excited at 730 nm and detected at 460 nm. The maximum projection of each image was obtained from a 5 μm-thick z-stack and analyzed with the CellProfiler program. Cells were delineated, and NADH fluorescent signal per cell was measured.

### 2.15. Detection of Lysosomes

After the corresponding treatments, cells plated in 35 mm dishes were incubated with 200 nM of Lysotracker red DND-99 (Invitrogen, L7528) for 5 min. Images were obtained with a two-photon microscope (LSM 710, Carl Zeiss, Germany) equipped with a Chameleon Ultra II pulsed infrared laser (Coherent Inc.), and a W-Apochromat 63x/1.0 M27 objective was used. Lysotracker was excited at 860 nm and detected at 590 nm, according to previous work [26]. The maximum projection of each image was obtained from a 5 μm-thick z-stack and analyzed with the CellProfiler program. NADH fluorescent signal was used to delineate the cell soma, and the number of lysotracker positive particles per cell was quantified.

### 2.16. Animals and Ketogenic Diet

Male CD1 mice (2.5 months old) were maintained at the animal facilities (certified by the Secretariat of Agriculture and Rural Development, SADER-Mexico) of Instituto de Fisiología Celular (IFC) at Universidad Nacional Autónoma de México (UNAM). Mice were handled following the *National Institute of Health Guide for the Care and Use of Laboratory Animals* (NIH publication No.8023 revised 1978), and experimental protocols were conducted under the current Mexican law for the use and care of laboratory animals (NOM-062-ZOO-1999) and approved by the local Committee for the Care and Use of Laboratory Animals (CICUAL, IFC-SCO174-21) of IFC. Only male mice were used in order to exclude possible effects of estrogens in female mice. A minimal number of animals was used.

Ten week-old mice were randomly distributed into two groups; one group was fed with control Chow diet (CD), and the second group was fed with a ketogenic diet (KD) (F3666, AIN-76A, BioServ, Canada, containing 75.1% fat; 8.6% protein, 3.2% carbohydrate, 4.8% fiber and 3.0% ash), *ad libitum* for 2 weeks. After 2 weeks of diet, animals were euthanized. The cortex and the hippocampus were extracted and homogenized in 1:10 weight/volume lysis buffer (50 mM Tris-HCl pH 8.0, 150 mM NaCl, 1% Triton X-100, 0.5% sodium deoxycholate, 1% SDS) containing 2 mg/mL of protease inhibitor cocktail.

### 2.17. Determination of β-Hydroxybutyrate and Glucose Levels in Blood

D-BHB and glucose were determined from blood samples taken from the tail vein (eight animals per group) using the blood glucose and ketone monitoring system (FreeStyle Optium Neo, Abbott Diabetes Care, Limited, Witney, Oxon, UK) and keto strips (FreeStyle Optium β-ketone). Data were collected from control diet and KD-fed animals before the onset of treatments and 2 weeks later when treatments were concluded.

### 2.18. Immunofluorescence in Brain Sections

After 2-week treatment with the CD or the KD, mice were anesthetized with sodium pentobarbital (0.1 mL/30 g weight) and perfused intracardially with 0.1 mM phosphate buffer/NaCl 0.9%. Brains were extracted, and the right hemispheres were transferred to 4% paraformaldehyde. After 3 days of fixation, brains were transferred to a 20–30% sucrose gradient for 24 and 72 h, respectively, and 40 μm floating coronal sections were obtained in a cryostat (Leica CM1510S). Permeabilization of brain sections was performed for 30 min in PBS/Triton X-100 0.9%. Sections were washed for 10 min in PBS and incubated in citrate 0.1% solution at 50 °C for 20 min. Afterward, brain sections were incubated for 15 min in PBS/glycine 0.1% at room temperature. Floating sections were blocked in PBS/BSA 2.5%/goat serum 2%/Tween 0.5%/Triton 0.9% at room temperature for 30 min. They were incubated with primary anti-LC3 (1:300, MBL International, PD014) and anti-SQSTM1/p62 (1:300, Abcam, ab56416) antibodies in PBS/BSA 2.5%/Triton X-100 0.3%/Tween-20 0.05%, during 48 h at 4 °C. Sections were washed 3 times 10 min each in PBS and incubated with appropriate secondary (Alexa 488 anti-rabbit or Alexa 594 anti-mouse, 1:300, 115.545.144 and 115.585.146 Jackson ImmunoResearch, respectively) in PBS/BSA 2.5% for 2 h. Sections were washed, and nuclei were stained with Hoechst 0.001% for 10 min. They were washed again, mounted on slides and covered with Fluoromont-GTM (Electron Microscopy Sciences, 17984-25). Confocal images were sequentially acquired with a Nikon A1R+ laser scanning confocal head coupled to an Eclipse Ti-E inverted microscope (Nikon Corporation, Tokyo, Japan) equipped with a motorized stage (TI-S-E, Nikon) and controlled through Nis Elements C v.5.00 software. Slides were analyzed under a PlanApo VC 60X WI DIC N2 water-immersion objective (N.A. 1.2); z-stack images were acquired every 500 nm using galvanometric scanners, both standard and GaAsP PMTs, excitation wavelengths of 405 (4.4 mW), 488 (1.4 mW) and 561 nm (4.8 mW) and pinhole aperture set at 33.21 μm.

### 2.19. Statistical Analysis

As indicated in figure legends, data are expressed as Mean ± SEM or as box-plot data showing median (line), 25th and 75th percentiles (bound of box) and minimum and maximum values (whiskers). Data were analyzed using Unpaired Student´s *t*-test, one-way ANOVA or two-way ANOVA followed by a Fisher multiple comparison test. Kruskal–Wallis test, followed by Dunn´s multiple comparison test, was used to compare mitochondrial area and maximal Feret´s diameter of mitochondria. A statistical significance of *p* ≤ 0.05 was used. All statistical analyses were performed using GraphPad Prism 7 software.

## 3. Results

### 3.1. D-BHB Stimulates Mitochondrial Function and Biogenesis

Mitochondrial function was determined in healthy cortical neurons maintained in normal glucose conditions (25 mM Neurobasal medium). At 8 DIV, they were treated with 5 mM D-BHB or L-BHB, the non-metabolic enantiomer of BHB, for 24 h. Changes in mitochondrial membrane potential were monitored using JC-1. Cells treated with D-BHB showed an increase in JC-1 aggregates (red fluorescence) and in the red/green fluorescence ratio relative to control neurons, indicating an increase in mitochondrial membrane potential. In contrast, L-BHB treatment induced no change (Figure 1A). D-BHB is known to increase the NAD^+^/NADH ratio [18]. Therefore, we used the autofluorescence properties of NADH when excited at 730 nm in a 2-photon microscope to follow the changes in NADH content [27,28]. We observed that in control neurons, NADH fluorescence is predominantly present in the cytoplasm; meanwhile, in the nucleus, there are low levels of NADH autofluorescence. When neurons were treated with D-BHB for 24–48 h, a significant decrease in the intensity of NADH autofluorescence was observed as compared to control neurons (Figure 1B), suggesting the increase in the NAD^+^/NADH ratio and the stimulation of the electron transport chain, as previously shown [18]. In cells treated with L-BHB, no change in NADH autofluorescence was found (Figure 1B). Together, these results indicate that the metabolically active enantiomer of BHB and D-BHB improves mitochondrial activity.

Then, we aimed to investigate whether D-BHB can promote mitochondrial biogenesis. Previous results have shown that the KD and BHB treatment can upregulate PGC1α, the master regulator of mitochondrial biogenesis in the brain [29,30]; however, it is still unknown whether it can be stimulated in healthy primary cultured neurons. First, the changes in *Pgc1α* mRNA were determined by qRT-PCR. As observed, the mRNA of this coactivator increased over time in neurons treated with D-BHB (Figure 2A). Accordingly, a corresponding significant increase in *Tfam* mRNA abundance (Figure 2B), a downstream gen of PGC1α, was also observed. Asignificant increase in the protein content of TOM20, a constitutive mitochondrial protein, was also observed suggesting increased mitochondrial mass (Figure 2C). These findings suggest the stimulation of mitochondrial biogenesis after 24–48 h exposure to D-BHB. To follow the changes in mitochondrial number and morphology in cells exposed to D-BHB, we observed mt-Keima, a fluorescent probe with a mitochondrial exporting signal sequence, by confocal microscopy. After treatment with D-BHB for 24–48 h, neurons exhibited higher mitochondrial area per cell (Figure 2D,E), an increased mean mitochondrial area and higher Feret´s diameter (Figure 2D,F,G), suggesting the presence of enlarged mitochondria. However, in cells treated with L-BHB, these changes were not observed (Appendix A). Altogether, these results indicate that D-BHB promotes mitochondrial biogenesis.

### 3.2. D-BHB Treatment Promotes Autophagy in Cortical Neurons

We have previously shown that D-BHB rescues the impaired autophagic flux in cortical neurons under glucose deprivation and glucose reintroduction (GD/GR) [10]. To test whether D-BHB stimulates autophagy induction in healthy neurons, we investigated the changes in the transformation of LC3-I to LC3-II as an index of autophagosome formation. As observed in Figure 3A, a significant increase in LC3-II levels was observed from 6–48 h in cells treated with D-BHB. SQSTM1/p62, a protein involved in autophagosomal cargo degradation, showed no changes (Figure 3B). To further corroborate autophagy induction, neurons were transfected with a plasmid encoding RFP-LC3 protein to detect autophagic vesicles. We observed a significant increase in the number of LC3-positive puncta in D-BHB-treated neurons at 24 h, confirming an increased formation of autophagosomes (Figure 3C). These results demonstrate that D-BHB can stimulate autophagy in healthy neurons. To assess whether D-BHB stimulates the autophagic flux, we blocked lysosomal degradation with chloroquine (CQ), a compound that increases the pH of lysosomes, preventing the function of lytic enzymes [31]. 

Whether D-BHB enhances autophagic degradation was first assessed by detecting the changes in LC3 and SQSTM1/p62 in the presence or absence of CQ. Using immunoblot, we observed that under control conditions, blocking lysosomal degradation for 4 h with CQ increases LC3-II levels (Figure 3D), indicating that there is a constant autophagosome formation in neurons. When neurons were incubated with D-BHB for 24 h, a clear elevation in LC3-II levels was observed, which increased further with CQ treatment (Figure 3D). This indicates that D-BHB not only augmented the formation of autophagosomes but also promoted autophagic vesicle degradation. The change in SQSTM1/p62 levels was also measured. A significant accumulation of this protein was not observed in the presence of CQ (Figure 3E), suggesting that in basal conditions, the degradation of this protein is low; however, incubation with D-BHB+CQ caused a significant accumulation of this protein (Figure 3E), confirming that D-BHB can stimulate basal autophagic degradation in healthy neurons.

Autophagy activation can be mediated by mTOR inhibition or AMPK activation. mTOR inhibition leads to dephosphorylation of ULK1 S757, inhibiting the autophagy initiation complex, while ULK1 S317 phosphorylation by AMPK activates autophagy initiation [5]. To test whether D-BHB stimulates autophagy initiation, we first investigated AMPK activation by determining its levels of phosphorylation at T172. pAMPK T172 was increased in cells treated with D-BHB, correlating with augmented pULK1 S317 (Figure 4A,B), suggesting that activation of AMPK is involved in the increased formation of autophagosomes induced by D-BHB. On the other hand, p-ULK1 S757 was not altered over time (Figure 4C), suggesting that autophagy induction by D-BHB is not mediated by mTOR signaling.

Lysosomes are essential organelles for autophagic flux, as the fusion of the autophagosome with the lysosome leads to cargo degradation through lysosomal lytic enzymes. Therefore, we investigated the changes in the number of lysosomes and lysosomal proteins in the presence of D-BHB. First, we evaluated the levels of LAMP2, a transmembrane lysosomal protein, and found that LAMP2 abundance increased in neurons treated with D-BHB from 6 to 12 h (Figure 5A). Changes in cathepsin B (CTSB) levels, a resident protease of the lysosome, were also determined. Immunoblot assay shows that D-BHB stimulates the production of the mature form of CTSB, a 25 kDa auto-proteolytic product of Pro-CTSB, from 12 to 24 h of treatment (Figure 5B), suggesting increased lysosomal activity. To detect lysosomes, we used lysotracker red and NADH autofluorescence to distinguish neuronal cytoplasm in living cells. An increased lysosomal number was observed in neurons treated with D-BHB after 24 h (Figure 5C). Together, the results suggest that D-BHB can stimulate lysosomal biogenesis, which is likely involved in the stimulation of the autophagic flux exerted by D-BHB.

### 3.3. D-BHB Treatment Promotes Mitophagy in Cortical Neurons

So far, results indicate that D-BHB increases autophagosome degradation, but it is unknown whether the degradation of specific components or organelles, such as mitochondria (mitophagy), is also increased. With this purpose, the degradation of mitochondrial proteins in the presence of CQ was determined to assess the mitophagic flux. Two mitochondrial proteins (TOM20, a transmembrane protein located in the outer membrane of the mitochondria, and COX IV, a member of the electron transport chain) were investigated. Under control conditions, treatment with CQ had no effect on TOM20 or COX IV abundance (Figure 6A,B), suggesting that these proteins are not degraded by the lysosomal pathway in control conditions. Consistent with mitochondrial biogenesis stimulation (Figure 2), when neurons were treated with D-BHB, a clear increase in the levels of TOM20 and COX IV was observed relative to the control without CQ. Importantly, when lysosomal degradation was inhibited with CQ, both TOM20 and COXIV abundance were significantly increased relative to D-BHB treatment alone (Figure 6A,B). These results suggest that D-BHB stimulates the lysosomal degradation of mitochondrial proteins.

To test whether D-BHB is indeed stimulating mitophagy, neurons were transfected with the RFP-LC3 plasmid to detect autophagic vesicles, and then cells were fixed and incubated with TOM20 antibody to detect mitochondria by immunofluorescence. Using confocal microscopy, we were able to detect autophagosomes colocalizing with mitochondria. Control neurons showed a low number of RFP-LC3 positive particles (autophagosomes), which increased when lysosomal degradation was blocked with CQ (Figure 6C,D). When neurons were incubated with D-BHB, a clear increase in the number of autophagosomes was observed relative to control, which further increased in the presence of CQ (Figure 6C,D), as previously observed by immunoblot (Figure 3D). The higher number of autophagosomes in Control+CQ-treated neurons did not correlate with a significant increase in the number and total area of TOM20-LC3 positive particles (Figure 6C,E,F). Interestingly, blocking the autophagic flux with CQ in D-BHB treated neurons increased not only the number of autophagosomes but also the number and total area of TOM20-LC3 colocalizing particles per cell (Figure 6C–F). These results indicate that the basal rate of mitophagy in neurons is low, as previously observed [32], which increases in the presence of D-BHB.

### 3.4. D-BHB Promotes Nuclear Translocation of Transcription Factors Essential for Autophagy and Lysosomal Biogenesis

Our results show that D-BHB can increase lysosome number and stimulate autophagy and mitophagy in healthy neurons; for this reason, we investigated the possibility that this ketone body can regulate the activity of master transcription factors involved in these processes, like FOXO1, FOXO3a and TFEB. First, to determine whether D-BHB induces the translocation to the nucleus of these factors, subcellular fractionations of cortical neurons treated with D-BHB were obtained. We observed that D-BHB increased the nuclear translocation of FOXO3a (Figure 7A) and FOXO1 (Figure 7B) from 6 to48 h, and from 12 to 48 h, respectively. These results indicate that D-BHB not only induces the expression of these transcription factors as previously reported [13,33] but also stimulates their redistribution to the nucleus in cortical neurons. Likewise, neurons treated with D-BHB exhibited a progressive increase in TFEB nuclear levels from 12 to 48 h of treatment (Figure 7C).

Then, using qRT-PCR, the expression of genes downstream these transcription factors [34,35,36,37] was determined. An increase in mRNA of lysosomal proteases (*Ctsb* and *Ctsd*) and lysosomal transmembrane proteins (*Lamp1* and *Lamp2*) was observed in neurons treated with D-BHB for 24 and 48 h (Figure 7D). In addition, a significant increase in the expression of autophagy genes such as *Atg5*, *Rab7*, *Map1lc3a* (LC3 protein gene) and *Sqstm1* (SQSTM1/p62 protein gene) was observed in the presence of D-BHB (Figure 7D). These results suggest that D-BHB is capable of promoting the expression of genes important for autophagy and lysosomal biogenesis by increasing the nuclear translocation of transcription factors such as FOXOs and TFEB.

### 3.5. SIRT2 Upregulation by D-BHB Mediates the Translocation of Transcription Factors Essential for Autophagy and Lysosomal Biogenesis

We next aimed to explore possible mechanisms to explain the induction of FOXOs and TFEB regulated by D-BHB. SIRT1 and SIRT2 are NAD^+^-dependent deacetylases that are involved in the regulation of autophagy; additionally, SIRT2 is highly expressed in the nervous system [38], as confirmed in cultured neurons using a specific antibody against SIRT2 (Figure 8A). Hence, the changes in the content of SIRT2 were followed by immunoblot after D-BHB exposure. SIRT2 shows three isoforms as a result of alternative splicing [38]. An increase in two of these three SIRT2 variants, SIRT2.1 and SIRT2.2, was observed after 12 to 48 h and 6 to 12 h exposure to D-BHB, respectively (Figure 8B).

SIRT2 can regulate TFEB levels [39] and the activity of FOXO1 and FOXO3a through their deacetylation [40,41]. Since we previously observed that D-BHB decreased NADH cellular content (Figure 1B), we hypothesized that this KB could modulate FOXOs and TFEB subcellular localization through SIRT2 activity. To test this possibility, we used AGK2, a SIRT2-specific inhibitor, and analyzed the abundance of FOXOs and TFEB in cytoplasmic and nuclear fractions (Figure 8C–E). As expected, AGK2 completely blocked the effect of D-BHB on FOXO1 and FOXO3a nuclear translocation (Figure 8C,D); surprisingly, it showed no effect on the localization of TFEB (Figure 8E). Consistently, inhibition of SIRT2 reduced the transcript levels of autophagy genes (*Map1lc3a* and *Sqstm1*) (Figure 8F), and the D-BHB-induced increase in LC3-II protein levels (Figure 8G). However, lysosomal genes (*Ctsb* and *Lamp2*) were not affected (Figure 8F). These results suggest that D-BHB regulation of FOXO1 and FOXO3a and the subsequent autophagy stimulation is mediated by SIRT2. In contrast, TFEB regulation by D-BHB is SIRT2 independent.

SIRT2 can stimulate autophagy and mitophagy [42], but also mitochondrial biogenesis through PGC1α in neurons [43], *Pgc1α* expression was tested in the presence of AGK2. As observed, AGK2 reduced both *Pgc1α* and *Tfam* expression (Figure 8F) in D-BHB-treated neurons, indicating that SIRT2 can regulate mitochondrial biogenesis. Moreover, SIRT2 inhibition diminished ATP levels in neurons treated and non-treated with D-BHB (Figure 8H). These results suggest that SIRT2 activity contributes to D-BHB effects on mitochondrial function and biogenesis.

Given that the D-BHB stimulation of TFEB is independent of SIRT2, we explored other possible mechanisms. It is known that TEFB activity can be negatively regulated by mTOR phosphorylation at Ser142 [35]. As observed in Figure 8I, TFEB phosphorylation at S142 decreases in the presence of D-BHB from 6–12 h and then partially recovers from 24–48 h, suggesting its possible regulation by mTOR rather than SIRT2.

### 3.6. Continuous Exposure to D-BHB Protects against Energy Stress-Induced Neuronal Death in a SIRT2-Dependent Manner

Neuroprotective effects of D-BHB have been observed when incubated during stressful conditions [8,10]; hence, we aimed to test whether D-BHB pre-treatment could also confer neuroprotection. With this purpose, neurons were pre-incubated for 24 or 48 h with D-BHB and then challenged with CCCP, a mitochondrial membrane uncoupler, for 24 h (Figure 9A). According to Figure 9B, when neurons were exposed to CCCP, a significant decrease in cellular viability was observed as assessed by the MTT reduction assay, which was partially recovered when neurons were pre-incubated with D-BHB. In agreement, CCCP induced an increase in LDH activity in the medium, indicating neuronal death, which was significantly reduced by D-BHB treatment (Figure 9C). In order to test D-BHB protection against other types of energy stress, neurons were exposed to glucose deprivation/reintroduction (GD/GR) (Figure 9D). Consistently, it was observed that treatment with D-BHB for 24 or 48 h significantly promoted cellular viability and decreased neuronal death-induced GD/GR (Figure 9E,F). Together, these results indicate that the pre-incubation of D-BHB confers resistance to damage induced by energy-limiting conditions.

Overexpression of SIRT2 in neurons has been observed to confer protection against oxidative damage [44]; therefore, we tested whether the D-BHB protective effect against GD/GR is mediated by SIRT2 activation. We observed that the SIRT2 inhibitor, AGK2, pre-incubated for 24 h before the exposure to GD/GR (Figure 9G), completely abated the protective effect of D-BHB (Figure 9H,I). Importantly, pre-incubation with the co-substrate of SIRT2, NAD^+^, exerted a similar protective effect to that of D-BHB (Figure 9J–L).

### 3.7. The Ketogenic Diet Upregulates FOXO1, FOXO3a, TFEB and PGC1α in the Mouse Brain

Finally, to confirm the present observations in an in vivo model, CD1 mice were treated for two weeks with a KD. First, a significant reduction in glucose and a significant elevation in D-BHB blood levels were observed in mice treated with the KD as compared to animals fed with a control diet (Figure 10A,B). 

Next, levels of transcription factors FOXO1, FOXO3a, TFEB and PGC1α were determined in different regions of the mice brains. A significant elevation in FOXO1 and FOXO3a content was observed in the cortex and the hippocampus of mice subjected to KD (Figure 10C–F); meanwhile, TFEB levels increased only in the hippocampus (Figure 11A,B). The PGC1α coactivator increased in both regions (Figure 11C,D). SIRT2 levels were also determined, and a significant increase was observed only in the hippocampus in animals fed with KD (Figure 11E,F). These results suggest that increasing the production of KB under normal conditions can stimulate transcription factors implicated in autophagy and lysosomal/mitochondrial biogenesis in the brain, possibly through the stimulation of SIRT2 activity.

### 3.8. The Ketogenic Diet Stimulates Autophagy and Mitophagy in the Mouse Brain

The possible activation of autophagy/mitophagy in the brain of mice treated with the KD was also investigated. As observed (Figure 12A,B), LC3-II significantly increased in the hippocampus of mice fed the KD, while in the cortex, a trend to increase was observed that did not reach statistical significance. SQTSM1/p62 showed a significant decrease in the cortex, while in the hippocampus, no statistical difference was observed (Figure 12C,D). LAMP2 abundance increased in both brain regions of mice of the KD group (Figure 12E,F). The changes in LC3 and SQTSM1/p62 were further explored by immunofluorescence in brain slices from mice treated with the CD and the KD. In agreement with immunoblot data, results suggest an increase in LC3 immunofluorescence in the cortex and the hippocampus of KD-fed mice. In addition, a decrease in SQTSM1/p62 immunofluorescence is suggested in the cortex (Appendix A). Together, the data suggest the stimulation of autophagy in the brain of mice treated with the KD, as previously reported [45].

Then, we aimed to investigate whether the KD induces changes in the content of the mitophagy proteins Parkin, BNIP3 and NIX/BNIP3L, which are regulated by FOXO1 and FOXO3a [21,36,46,47]. Parkin is an E3 ubiquitin ligase that is recruited from the cytosol to the outer mitochondrial membrane (OMM) by PINK1 when the mitochondrial membrane is depolarized and mitochondria are damaged. In mitochondria, Parkin promotes mitophagy by the ubiquitination of mitochondrial proteins leading to their degradation [48]. Parkin abundance was significantly increased in the cortex of mice fed the KD, while in the hippocampus, no significant elevation was found (Appendix A). BNIP3 and NIX/BNIP3L are mitophagy receptors located at the OMM. They bind to LC3, redirecting dysfunctional mitochondria to the autophagosome to be degraded. Dimerization of these proteins is involved in receptor activity and mitophagy initiation [49,50]; therefore, the dimeric/monomeric ratio of these proteins was determined in the brain of mice treated with CD or KD. The dimeric forms of BNIP3 and BNIP3L/NIX were significantly elevated in the cortex, while in the hippocampus, no significant change was observed (Appendix A). Altogether, these data suggest mitophagy activation in the cortex.

## 4. Discussion

The KD and the exogenous administration of D-BHB have recently been suggested as therapeutic tools against acute neurological disorders and neurodegenerative diseases [11]; however, the effects of ketone bodies on the nervous system under physiological conditions have not been completely elucidated. The present study shows that the continuous exposure of healthy cortical neurons to D-BHB improves mitochondrial metabolism and stimulates the basal rate of mitochondrial turnover through mitochondrial biogenesis and mitophagy. Importantly, these effects are associated with the upregulation of master transcription factors involved in autophagy in a SIRT2-dependent manner.

### 4.1. D-BHB Stimulate Mitochondrial Turnover

It is known that mitochondria require strict quality control to maintain energy balance within cells, particularly in neurons, due to their high energy requirements. Mechanisms involved in mitochondrial quality control include mitophagy, mitochondrial biogenesis, mitochondrial dynamics and the regulation of ROS production. When one or more of these mechanisms are impaired, mitochondria turn dysfunctional and can drive to neuronal death [51].

It has long been known that neuroprotection mediated by ketone bodies involves the improvement of mitochondrial function. Ketone bodies have been shown to reduce mitochondrial oxidative stress [52], improve mitochondrial respiration [53,54] and stimulate ATP generation [8,18]. Our results demonstrate that D-BHB induces mitochondrial biogenesis through PGC1α and TFAM expression in addition to improving mitochondrial function. Moreover, D-BHB shifted mitochondrial morphology to enlarged mitochondria and increased the mitochondrial area. However, at the same time, D-BHB promotes mitochondrial degradation by mitophagy, as observed by the increase in the colocalization of mitochondria with autophagosomes in D-BHB-treated neurons. This was associated with an increase in mitochondrial membrane potential and decreased NADH levels, suggesting the stimulation of mitochondrial respiration. Therefore, the present results suggest that D-BHB protective effects are related to the balance between the synthesis and degradation of mitochondria, improving both energy metabolism and neuronal health.

### 4.2. D-BHB Promotes the Autophagy–Lysosomal Pathway through the Regulation of Master Transcription Factors

The present results suggest that the continuous exposure of D-BHB to healthy neurons stimulates autophagy in an AMPK-dependent mTOR-independent manner, as D-BHB had no effect on ULK1 phosphorylation at S757 by mTOR, but increased AMPK activity (pAMPK T172) and pULK1 S357.This indicates the activation of the autophagy initiation complex. However, the mechanism responsible for AMPK activation remains to be elucidated.

D-BHB also stimulated the autophagic flux, as revealed by further accumulation of autophagic proteins and vesicles when lysosomal activity was blocked with CQ. Remarkably, mitochondrial proteins and LC3-TOM20 positive particles were also further accumulated after treatment with both D-BHB and CQ, suggesting that D-BHB stimulates mitochondrial degradation. Lysosomes are critical organelles for the completion of the autophagy–lysosome pathway, which is the rate-limiting step of mitophagy in neurons [32]. Lysosome number and activity are controlled by TFEB [35]. Results show that D-BHB increases TFEB nuclear levels, the expression of its downstream lysosomal genes and lysosomal number. TFEB is phosphorylated by mTOR at S142, S211 and S122, and the phosphorylation of S142 and S211 keeps TFEB inactive in the cytosol. mTOR inhibition by AMPK promotes TFEB dephosphorylation and its transport to the nucleus [55]. AMPK activity increased, while pTFEB S142 decreased after D-BHB exposure, suggesting the role of AMPK in TFEB regulation. The mechanism underlying TFEB regulation by D-BHB needs further study. Altogether, these results show, for the first time, that D-BHB continuous exposure stimulates TFEB in neurons, supporting that the enhanced autophagy/mitophagy flux is favored by TFEB-mediated lysosomal biogenesis.

### 4.3. SIRT2 Mediates Mitochondrial Turnover and Autophagy Regulation by D-BHB

FOXO1 and FOXO3a transcription factors can control autophagy and mitophagy [21,23,25], and their nuclear translocation can be regulated by SIRTs-mediated deacetylation [56]. A protective role of SIRT1 and SIRT3 against brain damage has previously been suggested [29,57,58,59], and both SIRTs are induced by the KD and BHB treatment in the brain and neurons [29,60,61]. SIRT2 is highly abundant in the brain of the adult mouse and is present in neurons and oligodendrocytes [38,62], as confirmed by our results showing that SIRT2 is present in the soma and neurites of cortical cultured neurons. The protective role of SIRT2 in the nervous system is still controversial. While its inhibition reduces ischemic injury [63] and α-synuclein toxicity [64,65,66,67], it also increases the antioxidant defense by FOXO3a, and its overexpression prevents oxidative stress-induced cell death in neuroblastoma [44]. The results shown here support that D-BHB upregulates SIRT2.1 and SIRT2.2 isoforms in neurons and that D-BHB’s protective effect against energy stress is mediated by SIRT2. Consistently, a similar protective effect was observed by NAD^+^, suggesting that an increase in the NAD^+^/NADH ratio by D-BHB is involved in SIRT2 activation.

SIRT2 can deacetylate FOXO1 and FOXO3a in adipocytes and kidneys [40,41], and hippocampal neurons from mice lacking SIRT2 show impaired autophagy/mitophagy [42]. However, the effects of the KD or D-BHB on SIRT2 activation and the control of autophagy have not been reported in the nervous system. Here, we report that D-BHB drives the nuclear translocation of FOXO1 and FOXO3a, in addition to the expression of their downstream autophagy genes. These effects were completely blocked by the SIRT2 inhibitor AGK2. Together, these results demonstrate, for the first time, that under physiological conditions, D-BHB upregulates FOXO1 and FOXO3a in a SIRT2-dependent manner. The KD and BHB exposure stimulates mitochondrial biogenesis through PGC1α [29,30]. Here, we show that SIRT2 is also implicated in mitochondrial biogenesis by D-BHB as the increased expression of *Pgc1α* and *Tfam* was blocked by AGK2. Furthermore, the inhibition of SIRT2 negatively affected ATP production in D-BHB- treated and non-treated neurons. Altogether, these results support the role of SIRT2 in mitochondrial quality control. Finally, our results demonstrate that two-week treatment with a KD elevates D-BHB blood concentration and upregulates FOXO1, FOXO3a, TFEB and PGC1α, coupled with increased SIRT2 abundance in the brains of mice. Moreover, results suggest that the KD also stimulated autophagy in the mouse brain, in agreement with a recent study showing that feeding mice with an enriched plant-origin fat diet can stimulate autophagy in the brain. The present results also suggest that a two-week KD stimulates mitophagy in the cerebral cortex. Little is known about the regulation of mitophagy in basal conditions, and further investigation is needed in order to elucidate the pathways involved in the regulation of mitophagy by the KD. Overall, the results suggest that the KD might promote mitochondria renewal, improving neuronal wellness in healthy individuals.

## 5. Conclusions

In conclusion, these findings are the first to report that D-BHB stimulation of the autophagic flux, under physiological conditions, improves mitochondrial quality control (Figure 13). Importantly, we identified SIRT2 as a target of D-BHB, which drives autophagy/mitophagy by the upregulation of FOXO1 and FOXO3a and mitochondrial biogenesis through PGC1α. Moreover, D-BHB promoted autophagy initiation by AMPK and lysosomal biogenesis through TFEB. All these actions render neurons less vulnerable to energy stress.

## Figures and Tables

**Figure 1 cells-12-00486-f001:**
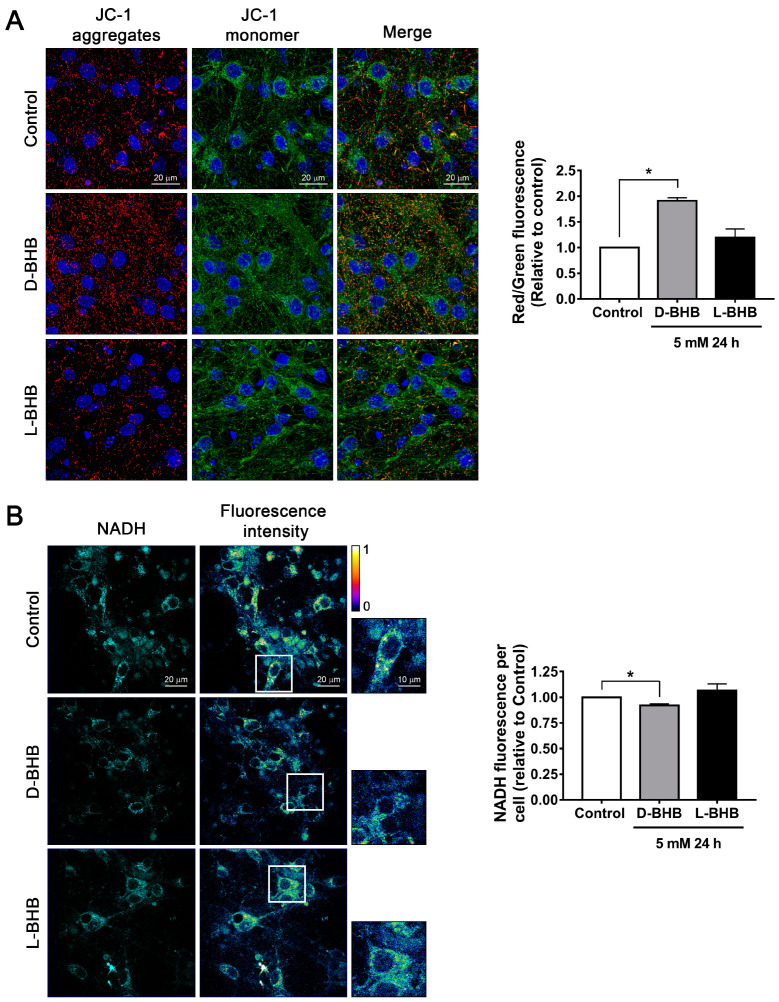
D-BHB treatment promotes mitochondrial function in cultured cortical neurons. (**A**) Left: representative images of mitochondrial membrane potential assay using JC-1 in neurons treated and non-treated with 5 mM of D-BHB or L-BHB for 24 h. Right: graph showing the red/green fluorescence ratio per condition shown as a fold change relative to control. (**B**) Left: representative images of NADH auto-fluorescence (cyan) and intensity spectrum (blue—lower intensity and yellow—higher intensity). Right: graph showing mean NADH intensity per cell of cortical neurons treated and non-treated with 5 mM of D-BHB or L-BHB for 24 h. Data are expressed as Mean ± SEM from 3 (**A**) or 5 (**B**) independent experiments. Data were analyzed by one-way ANOVA followed by Fisher’s post-hoc test. * *p* < 0.05 vs. control.

**Figure 2 cells-12-00486-f002:**
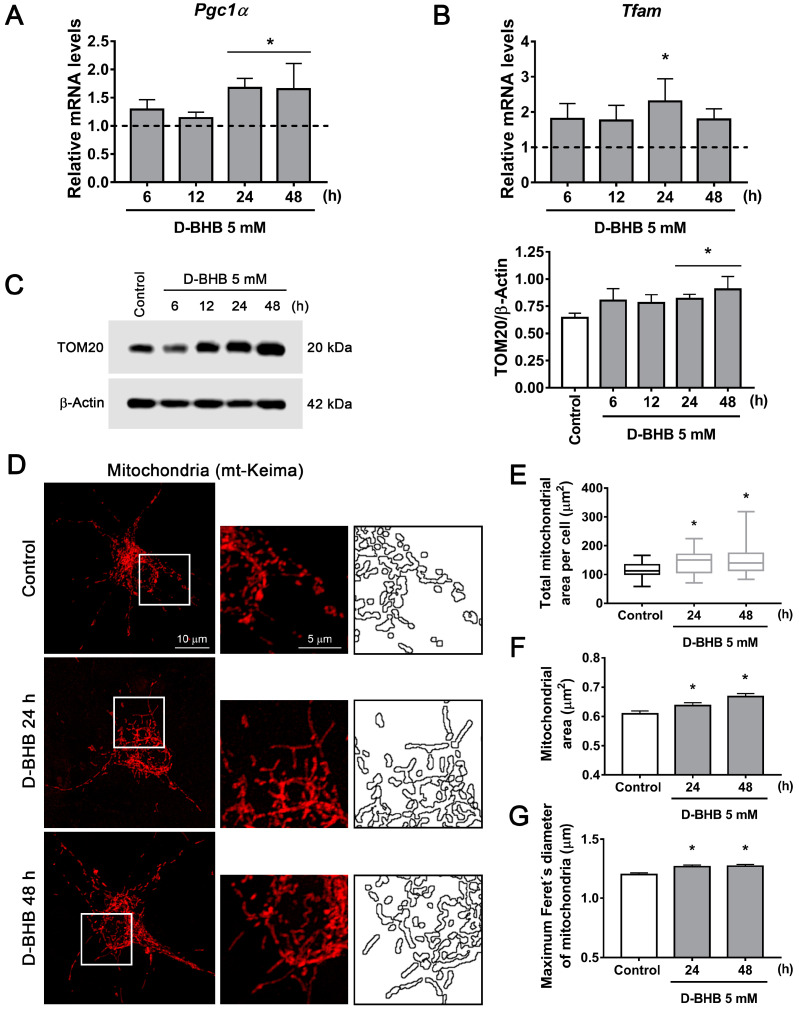
Effect of D-BHB on mitochondrial biogenesis in cortical neurons. Expression of (**A**) *Pgc1α* and (**B**) *Tfam* detected by qRT-PCR in control cultures and cultures exposed to 5 mM D-BHB for different times. (**C**) Representative Western blot and quantification of TOM20/β-actin. (**D**) Representative micrographs of neurons transfected with mKeima-red-mito (mt-Keima) and treated and non-treated with 5 mM B-BHB for 24 or 48 h. (**E**) Quantification of total mitochondrial area per cell, (**F**) individual mitochondrial area and (**G**) maximum Feret’s diameter of mitochondria treated and non-treated with 5 mM of D-BHB for 24 or 48 h. Data are expressed as Mean ± SEM from 4 (**A**,**B**), 6 (**C**) or 3 (**F**,**G**) independent experiments. Box-plot data (**E**) are expressed as median (line), 25th and 75th percentiles (bound of box) and minimum and maximum values (whiskers) from 23 cells of 4 independent experiments. Data from A, B and C were analyzed by one-way ANOVA followed by Fisher’s post-hoc test. Data from E, F and G were analyzed by Kruskal–Wallis test followed by Dunn´s post hoc test. * *p* < 0.05 vs. control.

**Figure 3 cells-12-00486-f003:**
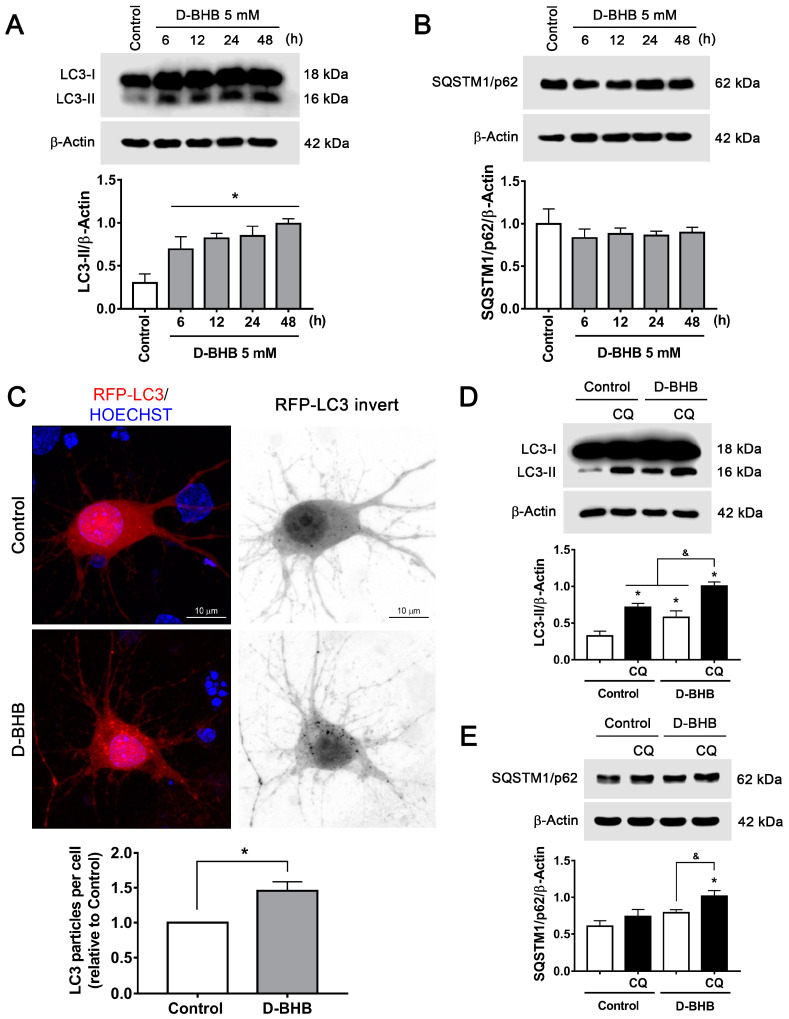
D-BHB induces autophagy in cortical neurons. (**A**) Representative Western blot and quantification of LC3-II/β-actin. (**B**) Representative Western blot and quantification of SQSTM1/p62/β-actin. (**C**) Representative micrographs of neurons transfected with RFP-LC3 and treated and non-treated with 5 mM B-BHB for 24 h. Graph shows quantification of RFP-LC3 positive autophagic vesicles in neurons treated and non-treated with D-BHB. Data were normalized with respect to control. Representative Western blot and quantification of (**D**) LC3-II/β-actin and (**E**) SQSTM1/p62/β-actin from cultures treated or non-treated with D-BHB (5 mM) for 24 h with or without chloroquine (CQ, 20 μM) for 4 h. Data are expressed as Mean ± SEM from 4 independent experiments. Data were analyzed by one-way ANOVA followed by Fisher’s post-hoc test for A and B or analyzed by two-way ANOVA followed by Fisher’s post-hoc test for D and E. * *p* < 0.05 vs. control; ^&^ *p* < 0.05 vs. D-BHB + CQ. Data in C were analyzed by Student´s *t*-test. * *p* < 0.05 vs. control.

**Figure 4 cells-12-00486-f004:**
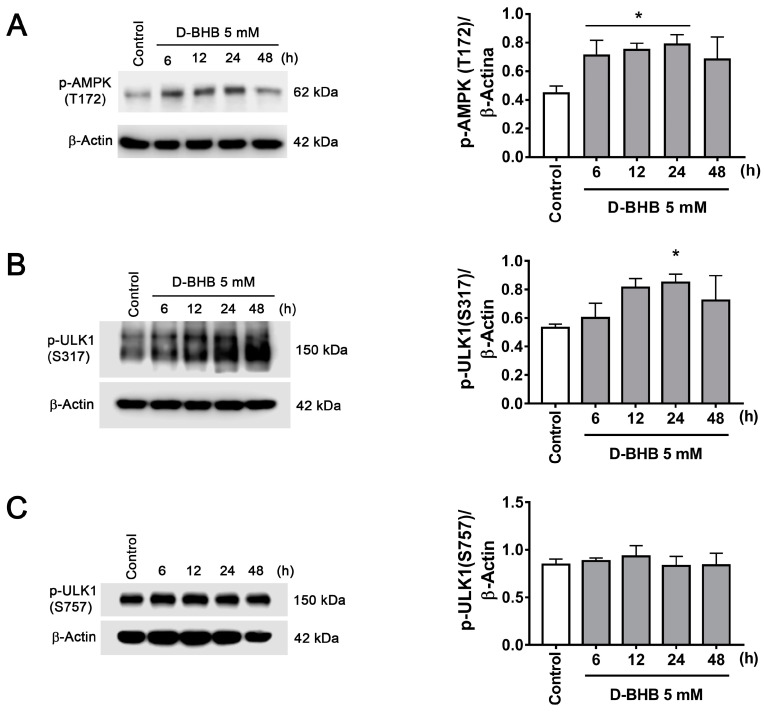
AMPK activation in neurons treated with D-BHB. Representative Western blot and quantification of (**A**) p-AMPK (T172)/β-actin, (**B**) p-ULK1 (S317)/β-actin and (**C**) p-ULK1 (S757)/β-actin from cortical neurons treated and non-treated with D-BHB (5 mM) for different times. Data are expressed as Mean ± SEM from 6 (**A**) or 4 (**B**,**C**) independent experiments. Data were analyzed by one-way ANOVA followed by Fisher’s post-hoc test. * *p* < 0.05 vs. control.

**Figure 5 cells-12-00486-f005:**
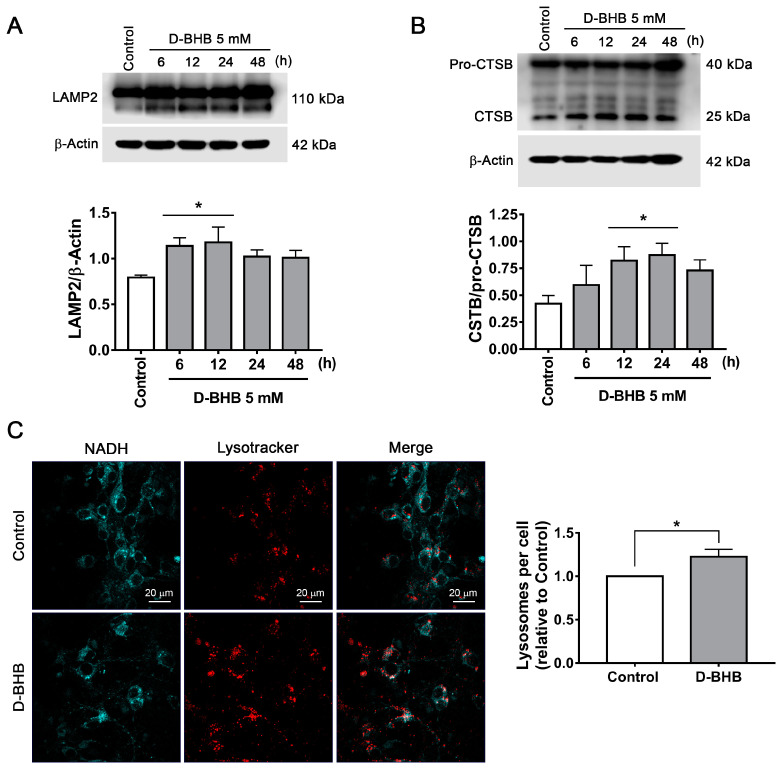
D-BHB promotes an increase in lysosomal proteins and lysosomal number. (**A**) Representative Western blot and quantification of LAMP2/β-actin and (**B**) CTSB/pro-CTSB. (**C**) Representative micrographs of NADH autofluorescence and lysotracker in neurons treated and non-treated with 5 mM D-BHB for 24 h. Graph shows quantification of lysosome particles in neurons treated or non-treated with D-BHB. Data are normalized with respect to control. Data are expressed as Mean ± SEM from 5 (**A**) or 4 (**B**,**C**) independent experiments. Data in (**A**,**B**) and were analyzed by one-way ANOVA followed by Fisher’s post-hoc test. * *p* < 0.05 vs. control. Data in C were analyzed by Student’s *t*-test. * *p* < 0.05 vs. control.

**Figure 6 cells-12-00486-f006:**
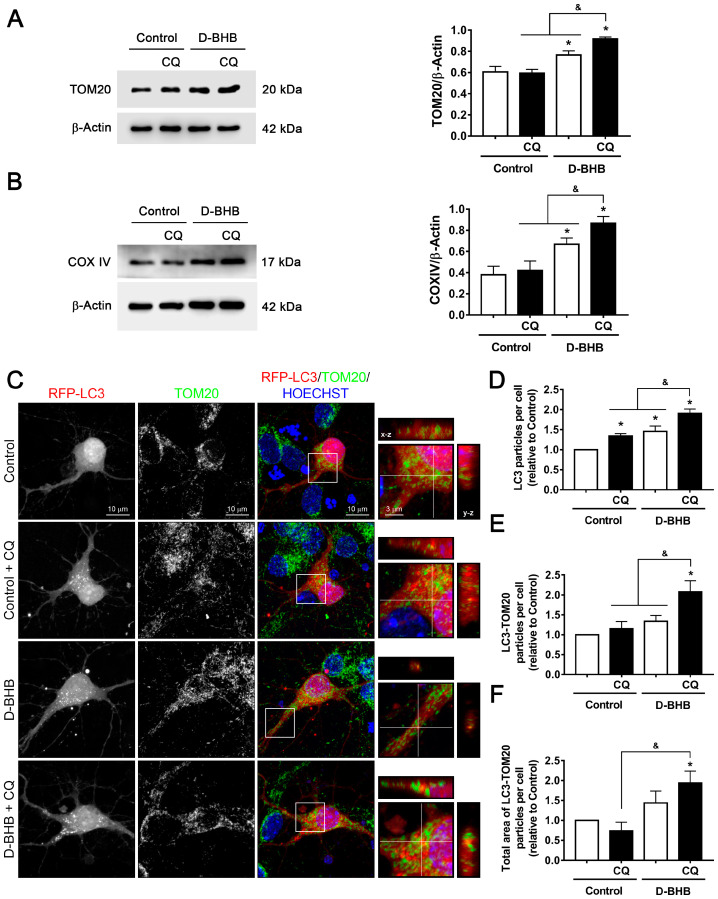
Effect of D-BHB on mitophagy in cortical neurons by colocalization of RFP-LC3 and TOM20. Representative Western blot and quantification of (**A**) TOM20/β-actin and (**B**) COXIV/β-actin from cortical neurons treated and non-treated with D-BHB (5 mM) for 24 h, with or without CQ (20 μM) for 4 h. (**C**) Representative images of neurons transfected with RFP-LC3 (red) and immunofluorescence against TOM20 (green), incubated or not with D-BHB (5 mM) for 24 h, and treated or non-treated with CQ (20 µM) for 4 h. Inserts show LC3-TOM20 positive particles in orthogonal y–z (right) and x–z (top) projections. Hoechst (blue) was used as nuclei marker. (**D**) Number of RFP-LC3-positive particles per cell. (**E**) Number of LC3-TOM20-positive particles per cell. (**F**) Total area of LC3-TOM20-positive particles per cell. Data are expressed as Mean ± SEM from 4 independent experiments. Data were analyzed by two-way ANOVA followed by Fisher’s post-hoc test. * *p* < 0.05 vs. control; ^&^ *p* < 0.05 vs. D-BHB + CQ.

**Figure 7 cells-12-00486-f007:**
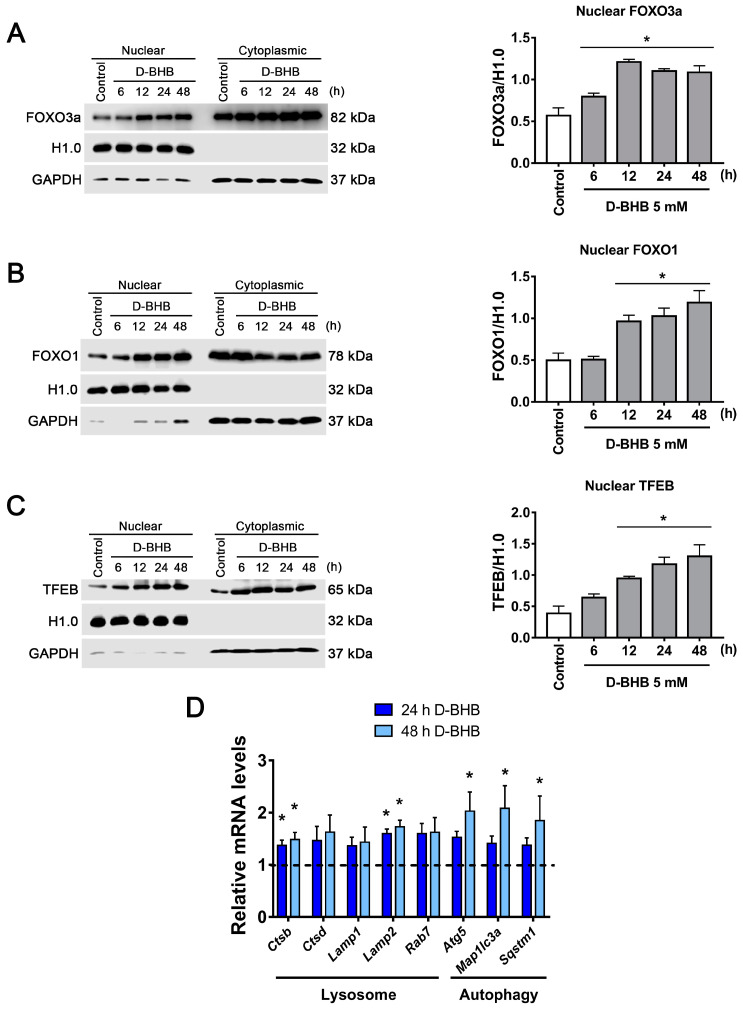
Nuclear localization of FOXOs and TFEB transcription factors in cortical neurons treated with D-BHB. Analysis of nuclear or cytoplasmic localization by subcellular fractionation of FOXO3a (**A**), FOXO1 (**B**) and TFEB (**C**). Representative immunoblot (left) and quantification (right) are shown. Histone H1.0 was used as a nuclear loading control, and GAPDH was used as a cytoplasmic loading control. (**D**) Expression of autophagy and lysosomal genes detected by qRT-PCR in cortical neurons exposed to 5 mM of D-BHB for 24 or 48 h. Data are expressed as Mean ± SEM of 4 (**A**–**C**) or 3 (**D**) independent experiments. Data were analyzed by one-way ANOVA followed by Fisher’s post-hoc test. * *p* < 0.05 vs. control.

**Figure 8 cells-12-00486-f008:**
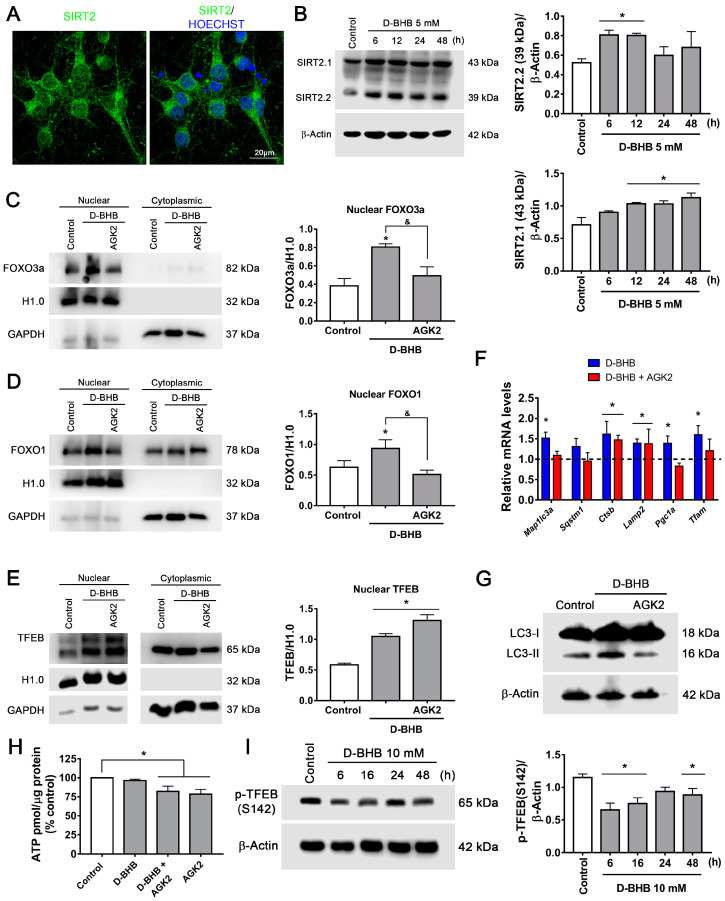
Effect of SIRT2 inhibition on FOXO3a, FOXO1 and TFEB nuclear localization in neurons treated with D-BHB. (**A**) Representative images of SIRT2 immunofluorescence in cortical neurons. (**B**) Representative western blot and quantification of SIRT2.1/β-actin and SIRT2.2/β-actin. Analysis of SIRT2 inhibition by AGK2 on nuclear or cytoplasmic localization of FOXO3a (**C**), FOXO1 (**D**) and TFEB (**E**). Representative immunoblot (left) and quantification (right). Histone H1.0 was used as a nuclear loading control, and GAPDH was used as a cytoplasmic loading control. (**F**) Expression of autophagy, lysosomal and mitochondrial biogenesis genes detected by qRT-PCR in cortical neurons exposed to 5 mM of D-BHB for 24 h with or without AGK2 (5 μM). (**G**) Transformation of LC3-I to LC3-II in cells treated with D-BHB with or without AGK2. (**H**) ATP levels in control cells and cells treated with D-BHB, AGK2 or AGK2+D-BHB. (**I**) p-TFEB (S142) levels in cells treated with D-BHB at different times. Representative immunoblot (left) and quantitative data (right). Data are expressed as Mean ± SEM from 3 (**B**), 6 (**C**,**E**) or 4 (**D**,**F**,**H**,**I**) independent experiments. Data were analyzed by one-way ANOVA followed by Fisher’s post-hoc test. * *p* < 0.05 vs control; ^&^
*p* < 0.05 vs. D-BHB.

**Figure 9 cells-12-00486-f009:**
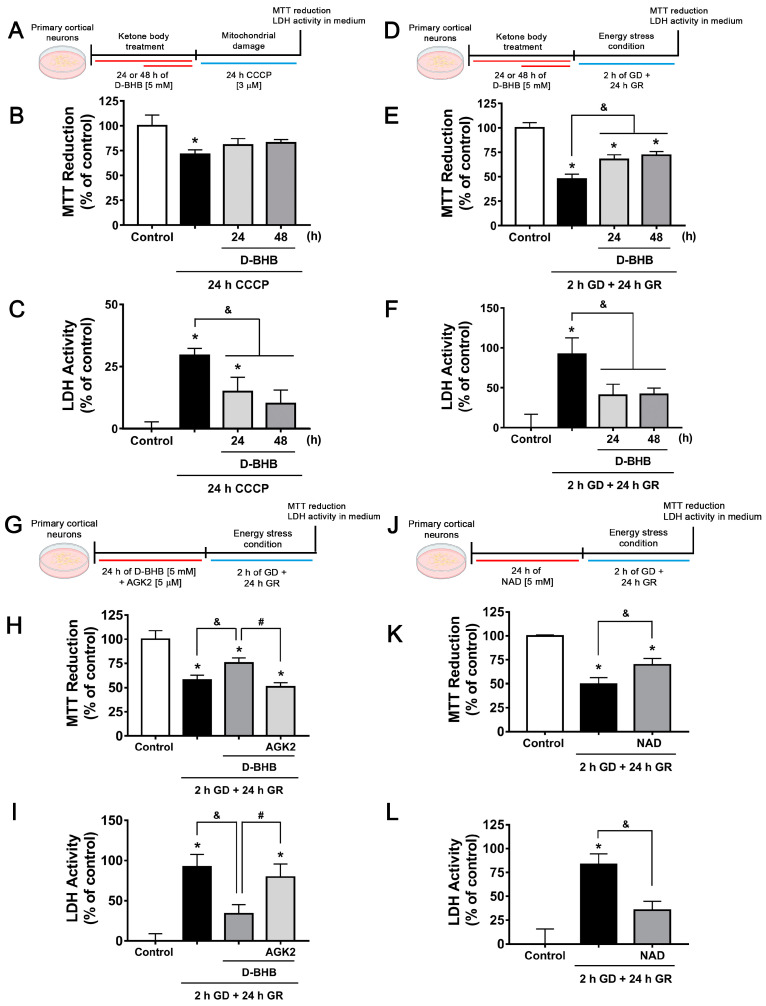
SIRT2 mediates the neuroprotective effect of D-BHB against energy stress. Diagram of the experimental design to test the neuroprotective effect of D-BHB pre-incubation using the mitochondrial uncoupler CCCP (**A**) or GD/GR in cortical neurons (**D**). MTT reduction (**B**,**E**) and LDH activity in culture medium (**C**,**F**) of cortical neurons pre-incubated 24 or 48 h with 5 mM D-BHB followed by 24-h exposure to 3 µM CCCP (**B**,**C**), or 2 h GD followed by 24 h GR (**E**,**F**). (**G**) Diagram of the experimental design to test the effect of SIRT2 inhibition by AGK2 on neuroprotection of D-BHB against energy stress induced by GD/GR in cortical neurons. (**H**) Reduction of MTT and (**I**) LDH activity of cortical neurons pre-incubated 24 h with 5 mM D-BHB with or without AGK2 (5 μM) followed by 2 h GD and 24 h GR. (**J**) Diagram of the experimental design to test the neuroprotective effect of NAD pre-incubation against GD/GR. (**K**) Reduction of MTT and (**L**) LDH activity. NAD (5 mM) was pre-incubated for 24 h followed by 2 h GD and 24 h GR. Data are expressed as Mean ± SEM of 3 (**B**,**C**), 6 (**E**,**F**,**H**,**I**) or 4 (**K**,**L**) independent experiments. Data were analyzed by one-way ANOVA followed by Fisher’s post-hoc test. * *p* < 0.05 vs. control, ^&^
*p* < 0.05 vs. GD/GR, ^#^
*p* < 0.05 vs. D-BHB.

**Figure 10 cells-12-00486-f010:**
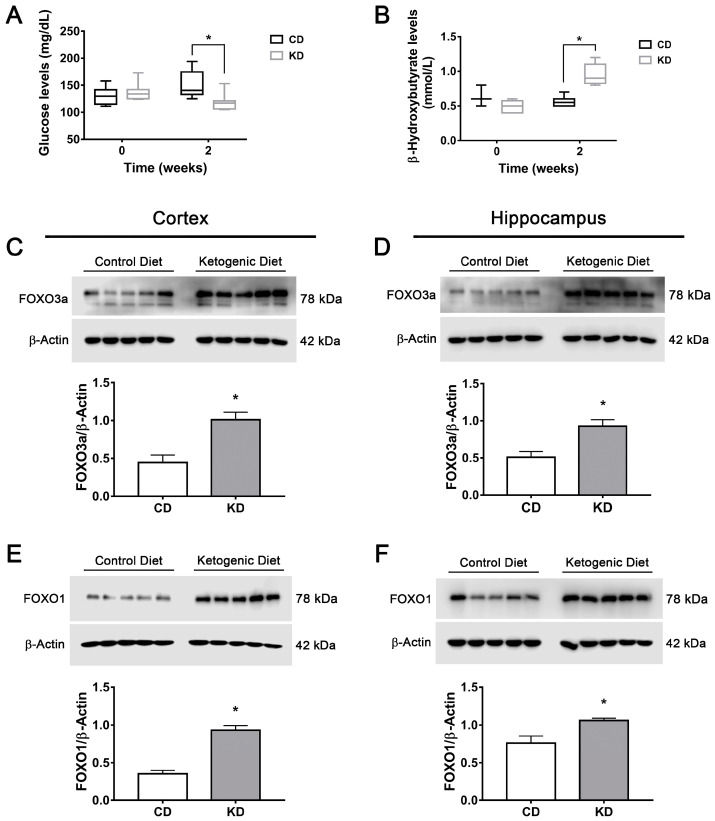
Changes in glucose, D-BHB and FOXO1 and FOXO3a levels in mice treated with a ketogenic diet. Glucose levels (**A**) and D-BHB levels (**B**) in blood of mice treated with a control diet (CD) or ketogenic diet (KD). Representative Western blots and quantification of FOXO3a/β-actin and FOXO1/β-actin in the cortex (**C**,**E**) and the hippocampus (**D**,**F**) of mice treated for 2 weeks with CD or KD. Data are expressed as Mean ± SEM from 8 individuals per group. Box-plot data (**A**,**B**) are expressed as median (line), 25th and 75th percentiles (bound of box) and minimum and maximum values (whiskers) from 8 individuals per group. Data from A and B were analyzed by two-way ANOVA followed by Tukey’s post-hoc test. Data from C to F were analyzed by Student’s *t*-test. * *p* < 0.05 vs. control diet.

**Figure 11 cells-12-00486-f011:**
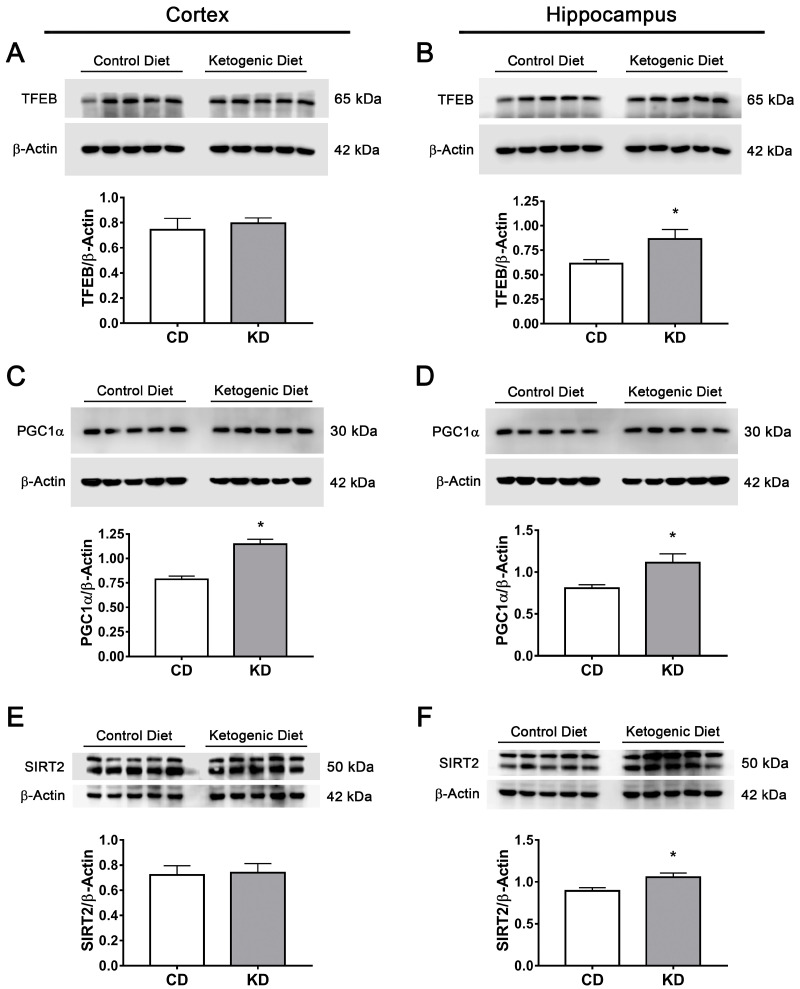
Changes in TFEB, PGC1a and SIRT2 brain levels of mice treated with control or ketogenic diet. Representative Western blot and quantification of (**A**,**B**) TFEB/β-actin, (**C**,**D**) PGC1α/β-actin and (**E**,**F**) SIRT2/β-actin in the cortex (**A**,**C**,**E**) and the hippocampus (**B**,**D**,**F**) of mice treated for 2 weeks with control diet or KD. Data are expressed as Mean ± SEM from 8 individuals per group. Data were analyzed by Student´s *t*-test. * *p* < 0.05 vs. control diet.

**Figure 12 cells-12-00486-f012:**
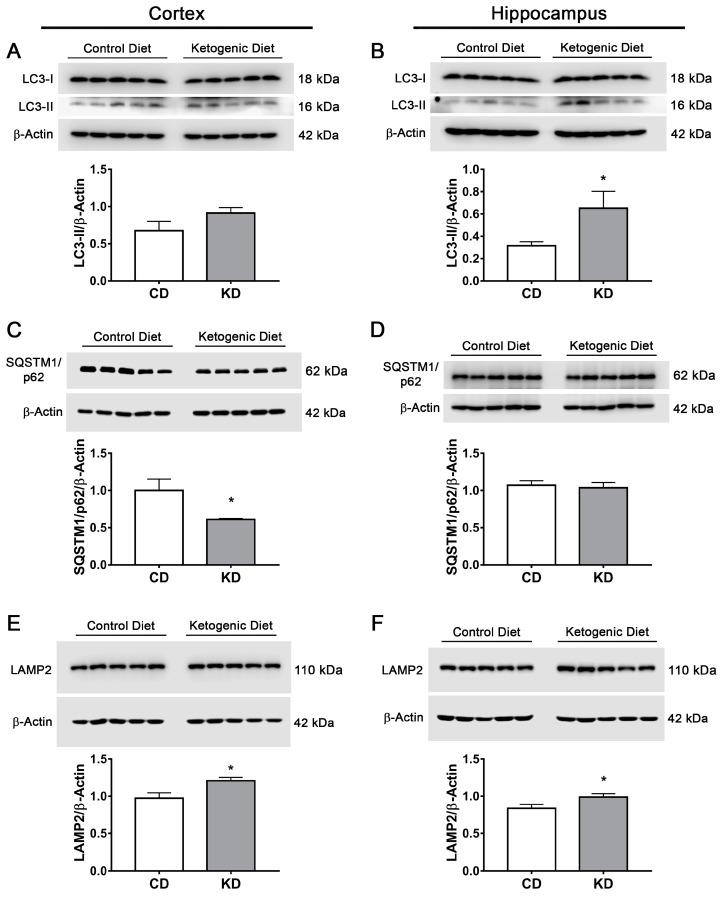
Changes in autophagy and lysosomal biogenesis markers in the brain of mice treated with control diet or ketogenic diet. Representative Western blot and quantification of (**A**,**B**) LC3-II/β-actin, (**C**,**D**) SQSTM1/p62/β-actin and (**E**,**F**) LAMP2/β-actin in the cortex (**A**,**C**,**E**) and the hippocampus (**B**,**D**,**F**) of mice treated for 2 weeks with control diet (CD) or ketogenic diet (KD). Data are expressed as Mean ± SEM from 8 individuals per group. Data were analyzed by Student’s *t*-test. * *p* < 0.05 vs. control diet.

**Figure 13 cells-12-00486-f013:**
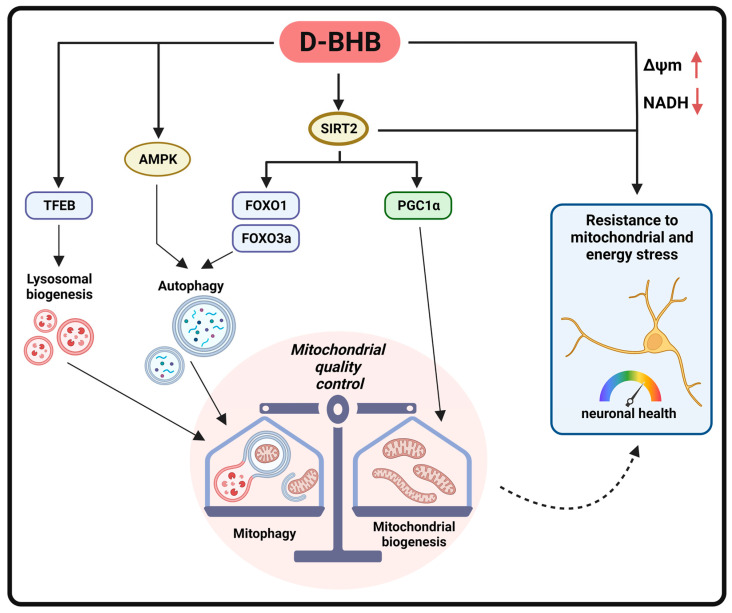
Schematic representation of the effects of D-BHB exposure to cortical neurons under physiological conditions.

## Data Availability

Data supporting the findings of this study are available from the corresponding author upon reasonable request.

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
