# Peer review of "Effect of the Ketone Body, D-β-Hydroxybutyrate, on Sirtuin2-Mediated Regulation of Mitochondrial Quality Control and the Autophagy–Lysosomal Pathway"

_cells, 2023, doi:10.3390/cells12030486_

Round 1

Reviewer 1 Report

The authors have done extensive work to study the role of ketone bodies in mitochondrial turnover. The study design is sound and the research methodology is well executed. However, there are few queries that needs to be addressed and a few minor edits to be done in the manuscript.

Line 245: ‘Lysotracker was excited at 860 nm and detected at 590 nm’. The excitation and emission maximum of Lysotracker red DND-99 is 577/590nm.

Results section-lines 283-285: Is it necessary to include the below paragraph in the manuscript? ‘This section may be divided by subheadings. It should provide a concise and precise description of the experimental results, their interpretation, as well as the experimental conclusions that can be drawn’

In figure 4B- the bar graph for p-ULK1 (S317)/β-actin, there is no significant difference between the control and 12hr data?

Line 328: Please correct the sentence ‘Then, south to investigate whether….’  

Did the authors study the levels/activity of the enzymes involved in the catabolism of ketone bodies like beta-hydroxybutyrate dehydrogenase and succinyl-CoA:3-ketoacid Coenzyme A transferase (SCOT) in the D-BHB treated neurons?

Adding a graphical representation of the proposed pathways by which ketone bodies regulate mitochondrial quality control and autophagy will give more insights and a better understanding of the mechanisms involved. 

Author Response

Reviewer 1

1. Line 245: ‘Lysotracker was excited at 860 nm and detected at 590 nm’. The excitation and emission maximum of Lysotracker red DND-99 is 577/590nm.

Many thanks for this observation. Indeed, the excitation and maximum emission of Lysotracker red DND-99 is 577/590 nm when using conventional confocal microscopy; however, in the present study, we detected Lysotracker in a 2-photon microscope, where, the excitation wavelength is different. Previous work has determined that the excitation of this compound in a 2-photon microscope can range from 1020 to 860 nm; therefore, we used 860 nm as the excitation wavelength. We have included a reference in the corresponding section to support the use of this particular excitation wavelength. 

2. Results section-lines 283-285: Is it necessary to include the below paragraph in the manuscript? ‘This section may be divided by subheadings. It should provide a concise and precise description of the experimental results, their interpretation, as well as the experimental conclusions that can be drawn’

This is a mistake. This paragraph should not have been included and has been removed.

3. In figure 4B- the bar graph for p-ULK1 (S317)/β-actin, there is no significant difference between the control and 12hr data?

Many thanks for the observation. When an ANOVA analysis is done with all groups no significant difference is found between the control and the 12 h time point. If the 12 h group is individually compared to the control condition by a Student’s t-test, then there is a significant difference between these groups (p=0.0074).

4. Line 328: Please correct the sentence ‘Then, south to investigate whether….’  

Thanks for the observation, the sentence has been corrected for “Then we aimed to investigate …..”

5. Did the authors study the levels/activity of the enzymes involved in the catabolism of ketone bodies like beta-hydroxybutyrate dehydrogenase and succinyl-CoA:3-ketoacid Coenzyme A transferase (SCOT) in the D-BHB treated neurons?

The regulation of ketolytic and ketogenic enzymes in the brain by the KD is an interesting subject that was not addressed in the present study. To our knowledge, there is no information about the regulation of ketolytic enzymes in neurons after long-term exposure to D-BHB. However, in an in vivo study, Liskiewicz and collaborators (Frontiers in Cellular Neuroscience 2021 Vol 15 article 733607) observed that a KD with a high content of animal-based fats upregulates BHB dehydrogenase1 and SCOT enzymes in the hippocampus, but not in the cortex of mice. Our results also show regional differences between cortical and hippocampal responses to KD exposure. Whether these changes can be explained by differential regional effects of the KD on ketolytic enzymes, is an interesting possibility that could be addressed in future studies.

On the other hand, in the present study, we addressed whether ketolysis of D-BHB could be relevant for the induction/activation of SIRT2, using the non-metabolic enantiomer of BHB, L-BHB, which is not likely to be converted to AcoA in cultured neurons, as the action of BHB dehydrogenase is selective for D-BHB. As such, L-BHB showed no effect in NADH content, making unlikely the stimulation of SIRT2. Based on these observations, we can suggest that D-BHB metabolism is important for the stimulation of autophagy and mitophagy mediated by this ketone body, although this possibility needs further investigation.

6. Adding a graphical representation of the proposed pathways by which ketone bodies regulate mitochondrial quality control and autophagy will give more insights and a better understanding of the mechanisms involved. 

Thank you for the observation. We have included a graphical abstract (Fig. 13). 

Reviewer 2 Report

Reviewer:

In this study, the author demonstrated that D-β-hydroxybutyrate (D-BHB) increased autophagy and mitophagy through enhancing FOXO1, FOXO3a, and PGC1α nuclear levels in a SIRT2-dependent. While the data are presented overall, there are several critical points that need the author’s attention.

Major specific points:

1.     In CD1 mice treated with a KD, the blood level of D-BHB was increased, and FOXO1, FOXO3a, TFEB, and other genes were upregulated. Will there be changes to mitophagy and autophagy in CD1 mice treated with a KD? The author must do the western blotting and staining in the cortex or hippocampus to confirm mitophagy and autophagy in CD1 mice treated with a KD.

Minor specific points:

1.     In fig. 2, the X-axis label, the (h) should be put on behind 48(Just like 48 (h)). The author should correct all of it for the whole paper.

2.      Some labeling is so confusing. For the control group, some used C as a labeling, and others used Ctrl as a labeling. Some are used to control labeling. Such as in fig. 2-5 and fig. 7-8. All labeling should be consistent!

3.     In fig.4A, the p-AMPK level decreased after D-BHB treatment for 48h. However, the quantification shows that the p-AMPK level increased. The author should run another blotting for updating.

4.     Fig. 6's comparison of the several groups is more confusing. When compared to which group, the vehicle with the D-BHB group's Tom20 protein level increased, as seen in Figure 6A? The author should make it clear by labeling it in figures.

5.     In fig. 8A, a scale bar must be added to the image.

Author Response

Reviewer 2

 Major specific points:

  1. In CD1 mice treated with a KD, the blood level of D-BHB was increased, and FOXO1, FOXO3a, TFEB, and other genes were upregulated. Will there be changes to mitophagy and autophagy in CD1 mice treated with a KD? The author must do the western blotting and staining in the cortex or hippocampus to confirm mitophagy and autophagy in CD1 mice treated with a KD.

The authors thank the reviewer for this important comment. As recommended, we determined the changes in autophagy and lysosomal proteins in the brain of mice treated with the CD and the KD. LC3-II significantly increased in the hippocampus of mice fed the KD, while in the cortex a trend to increase was observed, that did not reach statistical significance. SQTSM1/p62 showed a significant decrease only in the cortex while LAMP2 abundance increased in both brain regions of mice fed the KD, suggesting that in the present conditions, the autophagic flux is stimulated mainly in the cortex. These data have been included in new Fig. 12 and described in a new section (3.8) in the text. These results were supported by immunofluorescence analysis of LC3 and SQTSM1/p62 in brain slices of mice treated with CD and the KD (Supplementary Fig. 2). 

As we had no adequate antibodies available for the colocalization of LC3 with a mitochondrial marker to support the presence of mitophagy in brain slices, we determined the changes in the abundance of the mitophagy proteins, Parkin, BNIP3 and NIX/BNIP3L, which are regulated by FOXO1 and FOXO3a. Parkin abundance was significantly increased in the cortex of mice fed the KD, while in the hippocampus no significant elevation was found. These results have been included in Supplementary Fig. 3 and are described in a new section (3.8) in the text. It is known that the activity of the mitophagy receptors BNIP3 and NIX/BNIP3L is augmented when they form dimers, therefore, we determined the dimeric/monomeric ratio of these proteins in the brain of mice treated with CD or KD. The dimeric forms of BNIP3 and NIX/BNIP3L were significantly elevated in the cortex, while in the hippocampus, no significant change was observed. These data suggest mitophagy activation in the cortex.

Minor specific points:

  1. In fig. 2, the X-axis label, the (h) should be put on behind 48(Just like 48 (h)). The author should correct all of it for the whole paper.

Thank you for the observation. This has been corrected in all graphs.

  1. Some labeling is so confusing. For the control group, some used C as a labeling, and others used Ctrl as a labeling. Some are used to control labeling. Such as in fig. 2-5 and fig. 7-8. All labeling should be consistent!

Thank you for the observation. Labels have been changed to "Control" in all cases.

In fig.4A, the p-AMPK level decreased after D-BHB treatment for 48h. However, the quantification shows that the p-AMPK level increased. The author should run another blotting for updating.

Thank you for the observation. We have increased the sample size in order to verify the results. As depicted in Fig. 4A, we confirmed that at 48 h pAMPK increases relative to control, although these data do not reach statistical significance when analyzed by one-way ANOVA (p=0.058). We have replaced the representative gel in order to reflect the quantification more accurately. 

  1. 6's comparison of the several groups is more confusing. When compared to which group, the vehicle with the D-BHB group's Tom20 protein level increased, as seen in Figure 6A? The author should make it clear by labeling it in figures.

Thank you for the observation we have made changes to this figure to make comparisons more clear.

      5. In fig. 8A, a scale bar must be added to the image.

Many thanks for the observation, we have added the scale bar to the image.

Round 2

Reviewer 2 Report

The author already has addressed all my concerns.